Bat activity is related to habitat structure and time since prescribed fire in managed pine barrens in New England

Kay Natalie 1
Sadlon Amelia 1
Bakermans Marja H. mbakermans@wpi.edu 1 2
1 Department of Biology and Biotechnology, Worcester Polytechnic Institute , Worcester , MA , United States of America
2 Department of Integrative and Global Studies, Worcester Polytechnic Institute , Worcester , MA , United States of America
Li Dongming
Electronic publication date: 2023 Sep 12
Publication date: 2023
Volume: 11
Electronic Location ID: e15888
Received 2023 Apr 12; Accepted 2023 Jul 20
Copyright: ©2023 Kay et al.
Copyright year: 2023
Copyright holder: Kay et al.
License: This is an open access article distributed under the terms of the Creative Commons Attribution License, which permits unrestricted use, distribution, reproduction and adaptation in any medium and for any purpose provided that it is properly attributed. For attribution, the original author(s), title, publication source (PeerJ) and either DOI or URL of the article must be cited.
License URL: https://creativecommons.org/licenses/by/4.0/

Keywords: Acoustic monitoring, Barrens, Myotis, Pitch pine, Scrub oak, Restoration

Funding: The authors received no funding for this work.

==============================
Background

Several insectivorous bat species are found in New England, yet research on them is still scarce. Current research shows the ecological importance of bats due to their control of insect populations, but they are endangered by habitat loss and White Nose Syndrome, among other threats. Pine barrens are an uncommon ecosystem found in New England that supports other rare taxa and could be important for these bat species.

Methods

With hand-held audio recorders, we surveyed for bats in Montague Plains Wildlife Management Area in Massachusetts and Concord Pine Barrens in New Hampshire in June 2022. Our study objectives were to (1) describe the most common bat species and (2) compare bat activity across different habitat types at two managed pine barrens in New England. In particular, we examined bat activity related to habitat type (scrub oak, mature pitch pine, treated pitch pine, hardwood forest), habitat structure (i.e., canopy closure), time since prescribed fire, and path width. We analyzed our data through generalized linear modeling and logistic regression.

Results

Overall, we were able to measure the presence of five out of the nine total species found in the area, including the endangered Myotis lucifugus (little brown bat). We recorded 293 bat calls, with the majority of calls from big brown bats (71%). We found significant differences (p < 0.05) in bat activity in relation to time since prescribed fire and habitat structure. The index of bat activity was greatest in pitch pine and hardwood forests and lowest in scrub oak and treated pitch pine habitats. With preliminary data, we also found that silver-haired bat presence was influenced by habitat type, with more detections at survey points in hardwood forests.

Discussion

These findings demonstrate the importance of pine barrens as an ecosystem that supports bats in New England. According to the activity of bats in our study, closed canopy and mature pitch pine habitats may be prioritized in conservation efforts at managed barrens for bat species. Further research is recommended to better understand the relationship between prescribed fires, which are common in managed barrens, and bat activity.

Introduction

Bats are generally understudied and over-feared. Their reputation as disease carriers has hindered conservation efforts (Frick, Kingston & Flanders, 2020), and their nocturnal nature makes them a difficult taxonomic group to study (Villarroya-Villalba et al., 2021). Research has recently emerged on how bats respond to land management techniques, particularly prescribed fire and timber harvesting (Gotthardt, Kelley & Schwager, 2015; Austin et al., 2018; Steel et al., 2019; Divoll et al., 2021).

Despite the fear and suspicion surrounding bats, they are an incredibly important taxonomic group. Bats constitute one-fifth of mammalian diversity, act as pollinators, and control common flying insect pests (Steel et al., 2019). Habitat loss has contributed to bat population decline as urbanization increases and demand for resources increases timber harvesting and agriculture (Frick, Kingston & Flanders, 2020). Other dangers that are increasing bat mortality include wind turbines, powerlines, and radio towers (Huzzen, Hale & Bennett, 2020). The highly migratory hoary bat (Lasiurus cinereus) makes up roughly one-third of bat-related wind turbine fatalities and is a poorly censused species that is present in New England (Cornman et al., 2021). In addition, White Nose Syndrome (WNS), caused by the fungus Pseudogymnoascus destructans, has ravaged North American bat populations (Brooks , 2011). The disease easily spreads in cold, moist environments and has caused over a 90% population decline in cave-dwelling northern long-eared bats (Myotis septentrionalis), little brown bats (Myotis lucifugus), and tricolored bats (Perimyotis subflavus; Cheng et al., 2021).

Understanding the bat ecology in an area can be accomplished through acoustic data gathered from audio surveys (Gorresen et al., 2008; Blejwas, Lausen & Rhea-Fournier, 2014; Gotthardt, Kelley & Schwager, 2015; Wordley et al., 2015; Austin et al., 2018; Fraser et al., 2020; Taillie et al., 2021). Such techniques require less labor and expertise than physical captures in addition to being non-invasive, while still providing insight into the species and activity levels of the area. Additionally, Gomes, Appel & Barber (2020) found that mist-netting in tropical areas favored the capture of bats in Family Phyllostomidae when other Families of insectivorous bats are in the majority. While we are not in a tropical study site, there is likely still a bias in which bats are caught by mist nets. Although acoustic data does not lead to exact population numbers due to our inability to determine if one or multiple bats were making calls, it can be an inherent first step to characterizing bat populations with presence or absence information.

Nine bat species (all insectivorous) are native to New Hampshire and Massachusetts (Brooks & Ford, 2005): big brown bat (Eptesicus fuscus), silver-haired bat (Lasionycteris noctivagans), hoary bat, eastern red bat (Lasiurus borealis), little brown bat, eastern small-footed bat (Myotis leibii), northern long-eared bat, Indiana bat (Myotis sodalis), and tricolored bat. Big brown bats, silver-haired bats, hoary bats, and eastern red bats prefer to roost in mature or dead trees, along with building crawl spaces. All four species are migratory (Whitaker & Hamilton, 2019). The other bat species (little brown bat, eastern small-footed bat, northern long-eared bat, Indiana bat, and tricolored bat) are all endangered in Massachusetts and New Hampshire, largely due to their exposure to WNS as they usually overwinter in caves. However, the Indiana bat has not actually been recorded in Massachusetts since 1939 (Massachusetts Division of Fisheriesildlife, 2017). During the summer, these bats roost in foliage, tree cavities, buildings, and rock crevices (Massachusetts Division of Fisheriesildlife, 2017). Big brown bats generally start foraging 20 min after sundown, mainly in riparian or open areas, and along edge habitats (Wisconsin Department of Natural Resources, 2022). Northern long-eared bats often forage in “structurally complex habitat” sometimes over water, while little brown bats primarily forage near or over water (Massachusetts Division of Fisheries and Wildlife, 2019). Three of the species—big brown bats, silver-haired bats, and hoary bats—have distinct lower-frequency calls that differ from the calls of Myotis species. This can make it difficult to distinguish the three species from each other given the few acoustic nuances that differentiate them. However, their unique calls make them easily distinguishable from most other bat species (Blejwas, Lausen & Rhea-Fournier, 2014).

Pitch pine (Pinus rigida)–scrub oak (Quercus ilicifolia) barren habitat is a rare ecosystem across the New England landscape that is known to support high Lepidopteran species richness (Wagner, Nelson & Schweitzer, 2003; Grand et al., 2004) and may, in turn, support bat populations (Wickramasinghe et al., 2004; Leuenberger et al., 2016). This habitat type was formed by the recession of glaciers and resulting sand deposits, which led to a suitable environment for fire-adapted vegetation. Due to decreased wildfires from human intervention, pitch pine barrens have become overgrown, resulting in changes in ecosystem structure and function and increased wildfire risk (King et al., 2011). Pine barrens’ unique characteristics lend themselves to support rare plant and animal species, so prescribed burns and tree thinnings are used to manage the land, helping create a balance of pitch pine and scrub oak (Simmons & Hawthorne, 2014). For example, King et al. (2011) demonstrated that birds associated with shrublands and open forests, which are comparatively rare in highly forested areas like New England, were more prevalent in the managed zones, while birds associated with closed forests were more prevalent in the overgrown forests.

This study examined bat populations at two managed barren sites in New England, Montague Plains Wildlife Management Area (WMA) in western Massachusetts and Concord Pine Barrens in southern New Hampshire. Our study objectives were to (1) describe bat species most commonly found and (2) compare bat activity across habitat characteristics in these two managed pine barrens. In particular, we examined habitat type (i.e., scrub oak, mature pitch pine, treated pitch pine, hardwood forest), habitat structure (i.e., canopy closure), time since prescribed fire, and path width. We hypothesized that bats would be more active in mixed habitats, like treated pitch pine, that allow for places to roost and space to forage (Nelson & Gillam, 2017; Wisconsin Department of Natural Resources, 2022), and bats would be most active in areas that were recently burned due to elevated insect abundance (Campbell et al., 2018; Taillie et al., 2021).

Methods

Study area

Montague Plains Wildlife Management Area (WMA) is located in western Massachusetts within the town of Montague and is owned by the Massachusetts Division of Fisheries and Wildlife. It is over 1,500 acres of a mixture of habitats that resulted naturally from glacial sand deposits. The WMA is characterized by rare vegetation and animals; these include pitch pine–scrub oak barrens, the endangered spreading tick trefoil plant (Desmodium humifusum), several moth species (Lepidoptera), the eastern box turtle (Terrapene carolina carolina), and bird species like the eastern whip-poor-will (Antrostomus vociferus) and brown thrasher (Toxostoma rufum; Simmons & Hawthorne, 2014). The four main habitat types, as described in King et al. (2011), are treated pitch pine, pitch pine forest, scrub oak, and deciduous forest (Figs. 1A–1D, respectively). Montague Plains WMA is actively managed through tree thinning and prescribed burns, which help to reinforce the presence of the historically fire-adapted pitch pine–scrub oak barrens, since this successional habitat may otherwise grow into a pine/oak forest. The wildlife management area contains some undisturbed forested areas, including pine/oak and patches of deciduous forest.

Figure 1 Habitat types of (A) pitch pine, (B) treated pitch pine, (C) hardwood, and (D) scrub oak in two managed pine barrens, Montague Plains WMA (Massachusetts) and Concord Pine Barrens (New Hampshire).

Photos were taken by Amelia Sadlon at the time of study in 2022.

The Concord Pine Barrens are managed in a similar manner to the Montague Plains Wildlife Management Area. The site is located in southern New Hampshire in the city of Concord. The site is managed by New Hampshire Fish and Game for the main purpose of providing habitat for the endangered Karner blue butterfly (Lycaeides melissa samuelis). The pine barrens are maintained with prescribed fires and mechanical disturbance. Native plants, especially lupine (Lupinus), are also introduced back into the approximately 300-acre site. In addition to the pine barrens, young deciduous forest is present in non-disturbed areas (US Fish and Wildlife Service, 2003; Holman & Fuller, 2011). Access to the study sites was provided by staff at the NH Fish and Game Department (H. Holman, S. Houghton) and MA Division of Fisheries and Wildlife (A. Vitz).

Surveys

We surveyed four points (>300 m apart) per habitat type at Montague Plains WMA for a total of 12 points, and one point (80–230 m apart) per habitat type at the Concord Pine Barrens for a total of four points (Fig. 2; Kay, Sadlon & Bakermans, 2023). Across the month of June 2022, surveys were repeated 3–4 times at each survey point. We performed surveys on foot using the Echo Meter Touch 2 Pro (Wildlife Acoustics) connected to a cellular device. The settings for the Echo Meter Touch 2 Pro were as follows: the audio division ratio was 1/20, nightly sessions mode was off, save noise files and real-time auto-ID were both on, auto-ID sensitivity was balanced, trigger sensitivity was medium, the trigger window was 5 s, the maximum trigger length was 15 s, the gain was medium, and the sample rate was 256 kHz. We conducted surveys for 5 min at each survey point, similar to Kotowska et al. (2020), beginning between sunset until 40 min post-sunset, and varied the survey order each night to avoid temporal bias in sampling. This shorter sampling window allowed us to gather data at more points per night with limited devices available. All of the surveys took place on paths that ran along edge habitat, which helped to reduce noise interference from any overhead leaves and increased the likelihood of bat activity as supported by research showing a positive correlation with edge habitat (Nelson & Gillam, 2017; Taillie et al., 2021). Path width was recorded at each survey point at sites, using a range finder (Bushnell Yardage Pro Sport 450). Some conditions, like rain or heavy wind, interfered with our ability to accurately capture bat calls, so we chose evenings that minimized those conditions. On survey nights, we recorded the weather conditions and wind speed.

Figure 2 Map overview of survey locations at Montague WMA (Massachusetts, aerial imagery from summer 2019) and Concord Pine Barrens (southern New Hampshire; imagery from summer 2019).

Habitat and management determination

We determined habitat types at Montague Plains and Concord Pine Barrens by characterizing the dominant habitat within 28 m (i.e., the maximum distance that the Echo Meter is able to detect bats; (Ednie, Bird & Elliott, 2021)) surrounding each survey point. Habitat types were informed by King et al. (2011), including pitch pine, treated pitch pine, deciduous, and scrub oak (Fig. 1). Pitch pine habitats were characterized by a range of forest growth, including older hardwood trees (oak (Quercus), red maple (Acer rubrum)), white pine (Pinus strobus), scrub oak, and berry bushes (blueberry (Vaccinium sect. Cyanococcus) and huckleberry (Vaccinium membranaceum)), as well as oak and pine saplings. Treated pitch pine habitats contained fewer trees (i.e., <40% tree cover; King et al., 2011), with mainly pitch pine and occasional oaks present, as well as underbrush including scrub oak and blueberry bushes. Deciduous habitats featured a variety of trees, including oaks, red maples, and hickory trees (Carya), as well as pitch pine and white pine; however, the shrub layer was notably reduced, with a general absence of scrub oak that seemed to be replaced by tree saplings, mountain laurel (Kalmia latifolia), and black cherry (Prunus serotina). Scrub oak habitat was absent of most trees (i.e., <25% tree cover) and abundant with scrub oak and berry bushes (King et al., 2011).

Because bats may be more active in a habitat depending on its structure, such as a landscape with fewer trees and a greater open understory for space to fly and forage, we considered the structure of habitat as a potential factor influencing bat presence and activity (Nelson & Gillam, 2017; Wisconsin Department of Natural Resources, 2022). We categorized habitats as either open or closed canopy, where habitat types with <40% tree cover were “open” and >80% tree cover were “closed” (King et al., 2011). All survey points with treated pitch pine and scrub oak habitats were “open” and all with pitch pine habitats were “closed”; most survey points with a deciduous habitat type were “closed”, except for one “open” survey point at the Concord site.

Studies show relationships between insect abundance and prescribed fire activity, which could impact bat presence and activity (Campbell et al., 2018; Taillie et al., 2021), so we also collected data on prescribed burn activity at each survey point based on maps supplied by the Massachusetts Division of Fisheries and Wildlife, and NH Fish and Game. Four categories were created to describe the time since the last prescribed burn (Taillie et al., 2021; Braun de Torrez, Ober & McCleery, 2018). The “very recent” (VR) category contained areas burned in the last 0–5 years, “recent” (R) contained areas burned in the last 6–10 years, “distant” (D) describes areas burned in the past 10+ years, and “never” (N) describes areas that were never burned using prescribed fire (Kay, Sadlon & Bakermans, 2023).

Species determination

The Echo Meter Touch Bat Detector provides a real-time auto-identification functionality, which we used to determine bat species associated with survey points. Given some similarities in the frequency and spectrogram shape of certain bat calls (such as the calls of the big brown bat, silver-haired bat, and hoary bat), total certainty in the results from the Echo Meter results could not be guaranteed, so the data was reviewed manually as well (Maxell et al., 2015). We randomly chose 25% of recordings with a positive bat species ID to review manually, similar to Penone et al. (2018), who randomly reviewed 50 of their recordings. Additionally, some data were returned as “No ID” due to a weak signal; these files were all manually reviewed to ensure no bat call was present. If we determined a bat call was present, we manually identified the call using the guide from Maxell et al. (2015). We used the Kaleidoscope Pro Analysis software (Wildlife Acoustics Inc.) to process all data files for further viewing and analysis of sound spectrograms and metadata.

Analysis

We examined the presence or absence of bat species, as well as an activity level index of these species among habitats. We defined bat activity levels as the average number of identifications (IDs) aggregated across species at each survey point, with higher numbers of IDs considered to be a signifier of higher activity. First, we generated descriptive statistics of bat activity and presence by site, habitat (treated pitch pine, pitch pine, deciduous, scrub oak), habitat structure (open vs. closed canopy), and bat species. Because bat activity was an index and not necessarily a measure of abundance, we also tested if the presence of each species was related to habitat type and structure using logistic regression.

We used generalized linear modeling to test for differences between mean bat activity in relation to (1) habitat structure, (2) time since prescribed burn, and (3) path width. We did not include habitat type in the model because it was correlated with habitat structure (Fisher’s exact test, p = 0.004). No other predictor variables were correlated (all p > 0.05). All statistical analyses were run in R (version 4.2.1, 2022-06-23; R Development Core Team, 2022). Data from this study can be found at https://doi.org/10.5281/zenodo.7812126. Given our low sample size and the preliminary nature of this study, we set alpha = 0.10.

Results

Our study captured calls from five bat species found in Massachusetts and New Hampshire: big brown bats, silver-haired bats, hoary bats, eastern red bats, and little brown bats. In total, we recorded 187 bat calls in Montague Plains and 106 bat calls in Concord Pine Barrens (Table 1) across surveys (08–28 June 2022). Of the calls, the majority were big brown bats (70.7%), followed by hoary bats (11.6%), silver-haired bats (3.1%), eastern red bats (2.7%), and little brown bats (1.4%), and 10.6% of calls remained inconclusive from unknown species. After manually analyzing randomly sampled files with bat species IDs, only 8.7% showed possible indistinction between big brown bat and silver-haired bat calls. Detections were present at each survey point with averages ranging from 0.25 calls to 14.0 calls. The average wind speed on survey nights was 5.7 mph, and the average temperature was 70°F (Kay, Sadlon & Bakermans, 2023).

Table 1 Number of bat identifications from captured audio across species and habitats in Montague Plains WMA (Massachusetts) and Concord Pine Barrens (New Hampshire), 2022.

See Fig. 1 for habitat types.

	Bat species	
Site	Big brown	Hoary	Silver-haired	Eastern red	Little brown	Unknown	Total calls per habitat	
Montague	
Hardwood	47	0	4	6	1	8	66	
Treated pitch pine	25	3	1	0	1	3	33	
Pitch pine	56	5	0	1	2	1	65	
Scrub oak	8	11	1	1	0	2	23	
Concord	
Hardwood	19	0	1	0	0	4	24	
Treated pitch pine	7	0	0	0	0	5	12	
Pitch pine	38	0	1	0	0	3	42	
Scrub oak	7	15	1	0	0	5	28	
Total calls per species	207	34	9	8	4	31	Total calls: 293	

In our surveys of bat activity, we found significant relationships related to the predictor variables (F10,15 = 3.21, p = 0.055), where prescribed fire (Chi-sq = 10.9, p = 0.125) and canopy (Chi-sq = 2.8, p = 0.094; Table 2) were significant effects. Specifically, we found a significant decrease in bat activity at survey points with recent prescribed fire (7.4%, SE = 3.1, t = −2.3, p = 0.041) and never had prescribed fire (4.8%, SE = 1.9, t = −2.5, p = 0.030; Table 2) compared to survey points surrounded by distant burns. We recorded the greatest bat activity in the distant burn category (8.4 calls, SE = 1.7) compared to the very recent burn category, which recorded 3.5 (0.3 SE) average calls (Fig. 3).

Table 2 Regression results (e.g., parameter estimates) from predictor variables (habitat structure, prescribed fire, and path width) used in examining bat activity at two managed pine barrens in New England.

Model	Estimate	SE	t value	P (>—t—)	
Intercept	10.60	1.68	6.32	<0.001	
Canopy-open	−2.78	1.67	−1.68	0.125	
Fire-never	−4.76	1.89	−2.52	0.031	
Fire- recent	−7.37	3.14	−2.35	0.041	
Fire- very recent	−3.88	1.86	−2.08	0.064	
Width	−0.04	0.06	−0.77	0.457	

Figure 3 Bat activity index across four prescribed fire categories at managed pine barrens, Montague Plains WMA (Massachusetts) and Concord Pine Barrens (New Hampshire), June 2022.

Prescribed fire activities range from very recently burned (0–5 years post-disturbance) to never being burned.

On average, a survey point that had open canopy had a 2.7% decrease in bat activity compared to a point with closed canopy (Table 2). Across the survey points with an open canopy structure and closed canopy structure, we recorded 120 calls and 173 calls, respectively. Sampling points with closed canopy structures had, on average, 6.7 (1.5 SE) calls recorded while open canopy points had 3.9 (1.0 SE) calls (Fig. 4). Although habitat type was not included in the modeling (to avoid multicollinearity), we recorded 90 calls in hardwood habitat, 45 calls in treated pitch pine habitat, 107 calls in pitch pine habitat, and 51 calls in scrub oak habitat. Mean number of calls across each habitat type varied from 3.1 (0.9 SE) in treated pitch pine to 7.6 (2.5 SE) in pitch pine (Fig. 5). We recorded the highest number of calls at any one survey point in pitch pine habitat at both Montague and Concord study areas (at 9.25 and 14.0 calls, respectively).

Figure 4 Boxplots of bat activity index for habitat types that had open or closed canopy structure at Montague Plains WMA (Massachusetts) and Concord Pine Barrens (New Hampshire).

Canopy structure was considered open with <40% canopy cover or closed with >80% canopy cover.

Figure 5 Bat activity index (mean of identifications per survey point) across habitat types at Montague Plains WMA (Massachusetts) and Concord Pine Barrens (New Hampshire), surveyed June 2022.

For each group of data, boxplots provide the sample minimum (bottom line), lower quantile (25% percentile that splits the lowest 25% of the data; the bottom portion of the box), median (the middle value of all data; the line in the box), upper quantile (75 percentile; the upper portion of the box), sample maximum (the top line), and the data points (represented by different colors for each habitat).

Furthermore, model results indicated a non-significant decrease in bat activity for every 1% increase in path width (0.04%, SE = 0.06, t = −0.77, p = 0.457; Table 2). Lastly, logistic regression analyses revealed there were no significant differences in the presence or absence of each bat species by habitat type or structure (all p > 0.0142) except for silver-haired bat where habitat type was significantly related to that species’ presence (Chi-sq = 7.64, df = 3, p = 0.054). Silver-haired bats were detected at 100% (n = 4) of the points in hardwood forests, 50% of the points in scrub oak, and 25% of the points in pitch pine and treated pine.

Discussion

Few studies have examined bat presence and activity at managed pitch pine–scrub oak barrens, such as the sites in Montague Plains and Concord Barrens (Jackson & Schwager, 2012; Gotthardt, Kelley & Schwager, 2015; Taillie et al., 2021). These pine barrens offer a unique habitat that hosts a number of at-risk species, like spreading tick trefoil plant and lupine (Simmons & Hawthorne, 2014), as well as a wide array of Lepidopteran species, such as the Karner blue butterfly (Webb, 2010; Simmons & Hawthorne, 2014; Leuenberger et al., 2016). As a baseline study, our goal was to perform repeat surveys to ensure the audio capture of a range of species present and characterize their activity (Fraser et al., 2020). Any confusion in the auto-ID of these recordings was between silver-haired bats and big brown bats, but after manually checking 25% of our data, the margin of error was deemed to be acceptable (>90% were correct). The abundance of big brown bats is consistent with other literature, such as Nocera et al. (2019). As can be seen by our recordings of little brown bats, which are endangered across New England, the repeat survey method in addition to using stationary surveys with multiple points per location likely aided our observation of this rare species (Braun De Torrez et al., 2017; Deeley et al., 2021). Research shows that little brown bats often select edge and open habitats, which is common throughout managed pine barrens where tree thinnings and prescribed fires take place (Brooks, 2009; Nelson & Gillam, 2017). Additionally, Brooks (2009) found that little brown bats heavily prefer areas over water, something our survey points did not include. Though the use of prescribed fires creates desirable edge habitats for little brown bats, there is evidence to suggest it must be used cautiously as little brown bats avoid heavily burned areas (Jung, 2019).

Overall, our study found that bat activity increased with distant burns and decreased in areas with very recent burns or no burns. These results are at odds with Braun de Torrez, Ober & McCleery (2018) who found that recent burns increased bat activity in Florida, though our study had limited data. Additionally, given the small sample size of little brown bats that were detected and our aggregation of data between species, we cannot make any recommendations about the use of prescribed fire for little brown bat conservation, and we recommend more research be done in this area. We were also able to record each of the four migratory species that are more common (big brown bats, hoary bats, silver-haired bats, and eastern red bats), similar to Gotthardt, Kelley & Schwager (2015), Silvis, Gehrt & Williams (2016), and Austin et al. (2018).

Bat activity was related to forest structure, where activity was increased in closed canopy points (hardwood and pitch pine) compared to open canopy points (scrub oak and treated pitch pine). Divoll et al. (2021) found that the Indiana bat and northern long-eared bat were positively correlated with closed canopy habitats. They tended to roost in open canopy spaces, but had high incidences in forested areas likely for foraging, so their ideal habitat has a combination of both open and closed canopy spaces (Divoll et al., 2021). While different bat species were studied in Divoll et al. (2021), their results still concur with our findings that the studied bat species were detected more in closed canopy areas as they foraged. The bat species observed in each study exhibit similar foraging behavior; however, they tend to differ in their hibernation and roosting habits, aspects not studied in our work (Massachusetts Division of Fisheriesildlife, 2017). In addition to further study on the aforementioned aspects, a study of how bats use different canopy structures throughout the night, similar to Gomes, Appel & Barber (2020), could be an interesting extension to fully examine the impact of the different habits in managed pine barrens on bat activity.

Another study by Taillie et al. (2021) in south Florida showed that bat activity was more closely related to vegetation cover than prescribed fire activity. Smaller bats tended to be more active in closed and cluttered canopy areas than larger bats due to their increased mobility and compact size (Taillie et al., 2021). We also observed in our study that smaller bats (eastern red bats, little brown bats, and silver-haired bats) appeared to be most active in closed canopy habitats. Additionally, the logistic regression analyses we ran with species vs. habitat type showed that only silver-haired bats had a significant relationship to habitat type and were more active in hardwood forest, while a study from Arkansas found more bat activity in pine and mixed pine/hardwood forests (Perry, Saugey & Crump, 2010). Other studies have found smaller bat activity to be higher in more cluttered habitats specifically during foraging, compared to bigger bats, likely due to their size (Patriquin & Barclay, 2003; Sleep & Brigham, 2003). There is also evidence to suggest that bats use different adaptations to overcome competitive exclusion. Species adapted to faster and farther flight may utilize feeding grounds farther away from roosting sites while more agile fliers can outcompete other species in closer areas (Roeleke, Johannsen & Voigt, 2018). Overall, Taillie et al. (2021) found that the most bat activity was recorded in an intermediate canopy cover with greater woody understory. Closed canopy seemed to be too cluttered for bats to perform their hunting maneuvers and open canopy lacked abundant prey and cover from predators (Taillie et al., 2021). Finally, a study conducted by Wordley et al. (2015) showed that bats in Indian tea plantations performed better when forest fragments or shade coffee plants were present.

The management techniques performed at these sites could affect bat presence and activity for several reasons. Prescribed fires could affect the insect community and, therefore, insectivorous bat activity (Leuenberger et al., 2016; Nelson & Gillam, 2017). Insect activity often increases after burns; however, there is little long-term research on the effects of prescribed burns on the nocturnal insect species bats feed upon (Taillie et al., 2021). The fires could also create preferred roosting sites by creating snags and removing unstable trees (Braun de Torrez, Ober & McCleery, 2018). The tree thinning and clearing that takes place at these sites could provide more suitable edge habitat for the bats to forage (Taillie et al., 2021). Our results showed that prescribed fires did not positively affect bat activity, as areas that were burned 10+ years ago or were never burned hosted the most bat activity, while more recent burns (within 10 years) hosted less activity, which is supported by other research (Loeb & Blakey, 2021). However, it is important to note that we cannot definitively state that prescribed fire dampens bat activity as we did not have an even representation of burn categories or a large sample size.

Conclusions

This study demonstrated that restored pine barrens in New England can support an array of bat species. Research performed in managed pine barrens is important because there is currently a strong push to restore New England pine barrens due to their rare yet vital habitat structure and composition (Robertson et al., 2015). Our results are based on acoustic bat call IDs representing an index of bat activity, not the number of bats in the area. It is possible that IDs came from a few bats or the same bat calling multiple times. This is important to note as the results show how bat activity is related to habitat types, not bat population numbers related to habitat type. Stationary recording devices were out of the scope of our project, so we implemented sampling multiple times at the same sites in the research design. Manual audio recording using a smaller, mobile device has its advantages as it provides an opportunity in its accessibility (size, mobility, cost) for citizen science projects, which benefits the field of ecology as it expands the physical range of projects and complements professional field research (Dickinson, Zuckerberg & Bonter, 2010). This methodology also allows for real-time observations of weather and other environmental conditions that may not be made using a stationary device left overnight. Future studies looking at the relationship between bat species, especially little brown bats, and prescribed fires are recommended using similar methodologies with an increased number of surveys and a focus on prescribed fire.

We thank the Massachusetts Division of Fisheries and Wildlife and NH Fish and Game for access and use of the Montague study site, as well as Zoë Swartley for helping to collect data at this site. We would like to thank the Department of Biology and Biotechnology at WPI for student support. We also thank Heidi Holman and Sandi Houghton (NH Fish and Game) for information and resources for Concord Pine Barrens and Andrew Vitz (Massachusetts Division of Fisheries and Wildlife), for providing resources related to the Montague Plains Wildlife Management Area. We thank reviewers for their thoughtful feedback and efforts towards improving our manuscript.

Additional Information and Declarations

Competing Interests

Author Contributions

Data Availability

The authors declare there are no competing interests.

Natalie Kay conceived and designed the experiments, performed the experiments, analyzed the data, prepared figures and/or tables, authored or reviewed drafts of the article, and approved the final draft.

Amelia Sadlon conceived and designed the experiments, performed the experiments, analyzed the data, prepared figures and/or tables, authored or reviewed drafts of the article, and approved the final draft.

Marja H Bakermans conceived and designed the experiments, analyzed the data, prepared figures and/or tables, authored or reviewed drafts of the article, and approved the final draft.

The following information was supplied regarding data availability:

Data are available at Zenodo:

Kay, Natalie, Sadlon, Amelia, & Bakermans, Marja. (2023). Data from: Bat activity is related to habitat type and structure in managed pine barrens in New England [Data set]. Zenodo. https://doi.org/10.5281/zenodo.7812126.

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
