# Peer review of "Bat activity is related to habitat structure and time since prescribed fire in managed pine barrens in New England"

_PeerJ, doi:10.7717/peerj.15888_

## Round 0.1 · original submission · Major Revisions

This manuscript has been evaluated by three experts. They all found it is a nice work. However, they also address critical concerns regarding background information (introduction), specific methods, results, and discussion.

Reviewer 1 ·

Basic reporting

Well written

Experimental design

Design was mostly appropriate knowing the constraints of active sampling with little equipment. Might should look at a recent paper by Deeley et al. in the DC-area on active vs passive sampling in the post-WNS world. Also some work on that at Ft. Drum in NY some years ago. Basically so few bats on the landscape that while pre-WNS this was appropriate, it is problematic now.

As per analyses, unclear is the repeat samples were considered independent sample points or a repeated/nested measure. So your test with EPFU were appropriate but all the other bats I would convert to 0,1/presence absence and change the distribution to a binary/logistic response. Outcomes might change.

Validity of the findings

Largely OK. I disagree with MYLU foraging in complex structure. They are more an open water foraging species in the Northeast as confirmed by Brook's work at the nearby Quabbin and various studies in NY from Fort Drum to the Hudson River. Revise accordingly

Additional comments

no comment

·

Basic reporting

This manuscript is well written in a professional manner, and is structured appropriately. There is ample background information and appropriate references provided throughout. Raw data have been made available, however scrutiny of intermediate data and results would be made possible by inclusion of data manipulation and analysis scripts in the repository associated with this manuscript.

Experimental design

The research questions this manuscript aims to address are well laid out, as are the knowledge gaps it aims to fill. The data collection process was well documented, but there are ambiguities in the documentation of data processing and analysis that prevent replication of this research effort (see Validity of Findings).

Validity of the findings

There are substantial problems with unclear/missing raw data processing and analytical methods that make the validity of these findings impossible to assess, and the analysis impossible to reproduce without trial and error. The authors note that “some data were returned with no clear ID due to a weak signal; these files were manually reviewed to ensure a true bat call was detected or placed into an “unknown” species category” (l. 185–187) after mentioning that 25% of calls were reviewed manually for assessment of automated call identification accuracy. It is unclear whether the authors only reviewed 25% of the data for error quantification purposes, reviewed 25% of the data and used manual classifications obtained during this process in their analysis, or reviewed all noise/no species identification files for manual classification in addition to 25% of automated classifications with a positive species identification. The generally accepted means of assessing classifier accuracy is to review a portion of files that an automated classifier attributed a species identification to (no unidentified files are included in the stated proportion of files reviewed) for misclassification quantification. Revised or added classifications being used outside of classification error quantification is not an appropriate practice if only a portion of randomly selected files are being manually reviewed, as it would entail classifying parts of a dataset in different ways, and it is not clear if the authors did this. Manually inspecting and correcting species attributions for all files, or only files without a positive identification, would provide a consistent means of correcting and/or adding species attributions for calls in a manner that would not be liable to manually altering data in an inadvertently biased manner (i.e., proportionally more from one site than another).

The language surrounding the spatial hierarchy of data collection is also too vague for full understanding. With two broad areas (Montague Plains and Concord Pine Barrens), multiple intermediate areas (habitat types/structures within the broader areas) and multiple survey points nested within each of the intermediate areas, there must be consistent operational terms for each of these scales established. Because the authors used “site” as a presumed random intercept term (only specified as a random effect in the text), there are substantial implications associated with which spatial scale was used for this term, particularly for the coarsest spatial scale. Because there are only two broad areas (coarsest spatial resolution), it would generally be considered inappropriate to use this spatial scale as a random intercept due to insufficient factor levels (Harrison et al., PeerJ, 2018, 6). Without clarification regarding the spatial scale (“site” in the text without an operational definition for the term), I am unable to determine if the authors used a random intercept term appropriately.

Additionally, it is unclear why the authors elected to use post-hoc tests to compare three linear mixed effects models each containing only one fixed effect rather than fitting a single model with three fixed effects terms. The latter technique would allow comparison of coefficient estimates and test values between predictors generated by the same model. One may deduce that a more complex model structure led to convergence/collinearity issues, perhaps due to a relatively small sample size, but this is never explicitly stated. Some clarification regarding why the authors implemented what seems to be a somewhat unconventional analytical approach may support this choice.

Finally, it is unclear whether the authors aggregated call sequences by site (pooled across survey nights) or site-night for analysis. Aggregating call sequences by site-night would be most appropriate, as it preserves variation in the data that is lost through temporal aggregation. A hyperbolic example of variation lost through temporal pooling in the context of this research could be described by two sites, each with 100 call sequences detected over the course of four site-nights. If one site’s sequential nightly call sequence counts were all 25, and the other site had three nights with no detections and one night with 100 detections, those sites would look the same under a temporal pooling scenario but very different went aggregated by site-night. It is also worth noting that if the authors used the finest-scale definition (survey point) of “site” and aggregated by site-night, there would be no need for a random intercept term using site as there would be no spatial replication of data.

Additional comments

See annotated manuscript attached for more minor comments and suggestions. This largely consists of suggesting appropriate uses of en dashes and minor suggestions to improve clarity, but there are a few more substantive points worth bringing up here. While the authors allude to it in the discussion (l. 261–263), they never explicitly state that detections were aggregated across species. This is not necessarily problematic, though species-level, genus-level or sonotype aggregation is much more common for analysis of bat acoustic monitoring data (López-Bosch et al., Remote. Sens. Ecol., 2022, 8(2); Rowse et al., R. Soc. Open Sci., 2018, 5(6); Spoelstra et al., Proc. B, 2017, 284(1855)), and thus the aggregation of calls across species should be explicitly mentioned in the methods. Additionally, throughout the results (e.g., l. 226–229), the authors discuss differences in the data across factor levels, but do not discuss differences in coefficient estimates for each of the factors levels (generally relative to the first level alphabetically). While they do provide test values, reporting coefficient estimates or a derivative of them (e.g., the factor by which call sequences recorded changes when comparing one factor level to another) would base the results more squarely in the statistics than the data. Also, to reiterate a point mentioned in the Basic Reporting section, the Zenodo repository associated with this manuscript would benefit from the inclusion of the scripts used to prepare and analyze the data. While not strictly required, these would allow readers to clarify many of the ambiguities I presented above for themselves, and would substantially increase the reproducibility of this analysis.

·

Basic reporting

This is a nice, concise and straightforward study of bat activity using acoustic recording and identification. It was conducted in two differently-sized managed areas with readily identifiable habitat types. The manuscript is very well written and for the most part should be easy to understand even for international readers for whom English is a second language. I made a few minor comments directly on the pdf where a little grammatical correction could be made. The literature cited is relevant and appropriate. The figures for the most part are understandable (I made one small comment on the legend of Figure 3, asking for clarification of what the colored dots signify, which will also serve for figures 4 and 5.). The Table is also understandable. The basic data are provided through a web link, which is placed at the end of Methods, but should probably be placed in Results.

Experimental design

The methodology is clearly explained, fairly standard (and even popular for the current times) in its use of acoustic monitoring, which is highly recommended as a noninvasive method of detecting bat activity and automated identification, with human verification. The work is original, meaningful especially to bat conservation and land management for biodiversity, replicable, and the research question is clearly defined in both the Abstract and Introduction.

Validity of the findings

The authors are careful not to step outside the bounds of what their data actually tell them, and make it clear that they were not measuring things like species abundance, only relative abundance based on call IDs. The findings are interesting in showing the continuing use of managed habitats by the relatively more common bat species (expected from previous knowledge of species ranges).

Additional comments

I inserted Comments on the pdf version, many of which are simply questions that came to mind as I read the manuscript with interest. These are not meant to be criticisms or to require changes, but are only to get the authors to think about what a reader might ask about their study, as if listener questions in a conference presentation. However, in some cases, the comments might incite the authors to change or clarify their writing somewhat. The authors make an interesting comment in the Conclusions about an expanded potential future research stemming from this one, although not specifically stated as such. It involves the relative accessibility and ease of use of bat detectors by citizen scientists, who, if provided with additional detectors (with sufficient funding and a little training), could expand the authors’ data collection ability (given also appropriate land access) on these managed areas.

---

## Round 0.2 · accepted · Accept

The authors have addressed all the concerns of reviewers. As suggested by reviewers, the revised manuscript is much improved. It should be ready for publication.

·

Basic reporting

No comment

Experimental design

The changes made to the methods provide adequate detail for understanding and reproducing the work done in this paper.

Validity of the findings

Based on the authors' description of their analyses, the findings presented in this paper are adequately robust. I suggest the authors add their code used for analyses to the associated Zenodo repository for complete reproducibility, though if including data only is sufficient, this isn't a valid reason to request revisions.